# Leptin and PAI-1 Levels Are Decreased After a Dietary Intervention in Patients with Irritable Bowel Syndrome

**DOI:** 10.3390/ijms26104607

**Published:** 2025-05-11

**Authors:** Andreas-Balázs Szekely, Mohamed Nseir, Bodil Roth, Bodil Ohlsson

**Affiliations:** 1Department of Clinical Sciences, Lund University, 22100 Malmö, Sweden; an5824sz-s@student.lu.se (A.-B.S.); mo1138ns-s@student.lu.se (M.N.); bodil.roth@med.lu.se (B.R.); 2Department of Internal Medicine, Skåne University Hospital, 20502 Malmö, Sweden

**Keywords:** copeptin, C-peptide, insulin, IBS, leptin, low FODMAP, PAI-1, SSRD

## Abstract

A diet low in fermentable oligo-, di-, and monosaccharides and polyols (FODMAP) is an established treatment for irritable bowel syndrome (IBS). The starch- and sucrose-reduced diet (SSRD) is a newer, promising diet. Nutrients influence the production of gut hormones, which affect gastrointestinal motility. This study aimed to investigate the changes in copeptin, leptin, PAI-1, C-peptide, and insulin in patients with IBS following a 4-week dietary intervention and to explore whether these hormonal levels correlated with symptom improvements. A total of 142 participants with IBS were randomized to either a 4-week intervention with the SSRD (n = 70) or a low-FODMAP diet (n = 72). Participants completed the study questionnaire, food diary, ROME IV questionnaire, irritable bowel syndrome severity scoring system (IBS-SSS), and visual analog scale for irritable bowel syndrome (VAS-IBS) at baseline and after 4 weeks and 6 months; blood samples were collected at each of these time points. Leptin levels decreased from baseline to 4 weeks in the SSRD group (*p* = 0.002) but not in the low-FODMAP group (*p* = 0.153). In the overall study population, leptin (*p* = 0.001) and PAI-1 (*p* = 0.019) levels decreased from baseline to 4 weeks. Leptin changes were mainly associated with alterations in carbohydrate intake but not with symptom changes. These findings suggest that the SSRD reduces the levels of leptin in patients with IBS, while PAI-1 levels decrease independently of the dietary intervention, without a corresponding impact on symptoms.

## 1. Introduction

Irritable bowel syndrome (IBS) is a functional bowel disorder with a prevalence of 4.1% according to the ROME IV criteria [1,2]. Furthermore, approximately 40% of the population worldwide exhibit symptoms consistent with disorders of gut–brain interaction (DGBI) [2]. The pathophysiology of IBS is complex and multifactorial [3], and patients are frequently overweight, with a documented association between IBS, metabolic syndrome, and diabetes [4,5]. As patients with IBS often experience aggravated gastrointestinal (GI) symptoms in response to food intake [6], dietary modifications, such as those recommended by the National Institute for Health and Care Excellence (NICE) and a diet low in fermentable oligo-, di-, monosaccharides, and polyols (FODMAP), have been established as effective treatment strategies [7,8]. Since recent reports suggest that patients with IBS are predisposed to having reduced activity of the sucrase-isomaltase (SI) enzyme [9], the starch- and sucrose-reduced diet (SSRD) has recently been of interest in decreasing IBS symptoms [10,11]. FODMAPs and the malabsorption of disaccharides contribute to increased gas production and osmotic loads. This leads to water retention and secretion within the intestinal lumen, which causes luminal distension and symptoms [12,13,14].

Diet and nutrient intake influence the production of gut hormones regulating glucose homeostasis, gut motility, and satiety signals. Therefore, these hormones are of particular interest in IBS, as they may contribute to the pathogenesis and symptom burden. Furthermore, altered hormonal levels may have negative health effects and be related to other associated conditions, such as diabetes, metabolic syndrome, and overweight.

Glucose homeostasis is regulated by hormones such as insulin, C-peptide, gastric inhibitory peptide (GIP), glucagon, and glucagon-like peptide-1 (GLP-1) [15,16]. Following food intake, the plasma levels of GIP and GLP-1 increase rapidly, stimulating insulin biosynthesis and secretion from pancreatic β cells. Furthermore, these gut hormones play an important role in regulating gut motility, nutrient digestion, and intestinal absorption [16]. Although C-peptide is a prohormone of insulin and was previously considered biologically inert, recent evidence suggests that C-peptide administration in patients with diabetes improves certain physiological functions [15].

Plasminogen activator inhibitor-1 (PAI-1) is an adipokine mainly secreted by the adipose tissue, functioning as a plasma inhibitor of urokinase plasminogen activators (uPA), thereby regulating the fibrinolytic systems [17]. PAI-1 is also involved in the development of tumorigenesis, obesity, metabolic syndromes, diabetes, inflammation, and autoimmunity [18].

C-terminal pro arginine-vasopressin, commonly known as copeptin, is secreted in equimolar amounts with arginine-vasopressin (AVP) [19]. AVP plays an important role in the regulation of osmosis, body fluid balance, and glucose homeostasis. Previous studies have shown a correlation between elevated levels of copeptin and an increased risk of developing diabetes mellitus [20]. This increased risk may be influenced by the body mass index (BMI) [21]. Additionally, fructose intake may affect the release of copeptin [22]. There are no previous studies investigating copeptin concentrations in patients with IBS, highlighting a gap in the current understanding of the hormonal landscape in IBS.

Leptin plays a crucial role in the regulation of energy homeostasis by modulating appetite and food intake [23]. Leptin is mainly produced by the adipose tissue, but its presence has also been found in other tissue types, including the placenta and gastric chief cells in the stomach [24]. Leptin levels are directly proportional to the body fat mass, ranging from 5 to 10 ng/mL in healthy individuals to 40 to 100 in obese individuals [23,25,26]. Leptin levels rise transiently following food intake and decrease during fasting [27,28]. In the hypothalamus, leptin enhances short-acting peripheral saturation signals, such as the effects mediated by cholecystokinin (CCK) [29]. Peripherally, leptin enhances the release of GLP-1 from L-cells located in the distal ileum and heightens the vagal-afferent sensitivity to gut peptides [30]. These gut peptides also affect gut motility, nutrient digestion, and absorption [16].

It is well established that dietary modifications promote weight loss and alleviate symptoms in patients with IBS [7,8,10,11]. Previous studies have demonstrated decreased circulating levels of leptin, PAI-1, C-peptide, and insulin, without significant changes in GIP and GLP-1, following a 4-week SSRD intervention [31,32]. Symptom improvement following dietary changes may result from interactions between the luminal content, gut endocrine cells mediating hormonal release, and weight loss [5,30,33,34].

The first hypothesis of this study was that dietary interventions designed to alleviate IBS symptoms would also affect the concentrations of gut hormones that regulate glucose metabolism and appetite, consistent with findings from previous research [31,32]. Due to the hormonal effects on GI motility [16], the second hypothesis proposed that changes in hormonal concentrations would correlate with symptom alleviation.

The primary aim of the present study was to further investigate and confirm whether the concentrations of copeptin, leptin, PAI-1, C-peptide, and insulin would change before and after a 4-week dietary intervention with either the SSRD or a low-FODMAP diet in patients with IBS. Additionally, this study secondarily aimed to explore whether these hormonal changes could provide insights into the underlying mechanisms responsible for symptom alleviation and whether the hormonal effects remained at follow-up.

## 2. Results

### 2.1. Basal Characteristics

In total, 142 participants were included in the study; 70 participants were randomized to the SSRD group and 72 to the low-FODMAP group. The distribution of sex, age, and disease duration did not differ between the groups (Table 1). At baseline, 25 participants (17.6%) had constipation-predominated IBS (IBS-C), 42 (29.6%) had diarrhea-predominated IBS (IBS-D), 48 (33.8%) had mixed IBS (IBS-M), 6 (4.2%) had unspecified IBS (IBS-U), and 21 (14.8%) had functional bowel disorders (FBD) (pain once a week with less than 30% association with bowel habits). The distribution of IBS subtypes did not differ between the groups at any time point (Appendix A). The most common comorbidities and drug treatments are presented in Appendix A.

### 2.2. Dietary Changes and Weight

At baseline, starch intake was slightly higher in the low-FODMAP group compared to the SSRD group (*p* = 0.045) (Appendix A). After 4 weeks, both groups exhibited a significant reduction in body weight and kilocalorie intake (*p* < 0.001 for all). There was a significant correlation between weight reduction and calorie reduction (Rs = 0.196, *p* = 0.022). Both groups had significantly lowered intake of carbohydrates (*p* < 0.001), with lower intake in the SSRD group compared to the low-FODMAP group (*p* < 0.001). Protein intake increased in the SSRD group (*p* = 0.003), with a significant difference in intake between the groups (*p* = 0.017). Monosaccharide intake (*p* < 0.001 and *p* = 0.002, respectively) and sucrose intake (*p* < 0.001 and *p* = 0.027, respectively) decreased in both groups, although the decrease in sucrose was most pronounced in the SSRD group (*p* < 0.001). The intake of disaccharides, starch, and added sugar decreased only in the SSRD group, resulting in a difference between the groups (*p* < 0.001 for all), and these intakes remained reduced after 6 months (*p* = 0.027, *p* = 0.013, and *p* = 0.024, respectively). After 6 months, starch intake was also reduced in the low-FODMAP group (*p* = 0.008). Fiber intake decreased in the low-FODMAP group at both week 4 (*p* = 0.002) and at the 6-month follow-up (*p* < 0.001) (Appendix A).

### 2.3. IBS Symptoms

At baseline, no significant differences were observed between the groups regarding GI and extraintestinal symptoms (Appendix A). Both groups had a significantly reduced degree of specific GI symptoms [35], total scores on the irritable bowel syndrome severity scoring system (IBS-SSS), and total extraintestinal IBS-SSS at 4 weeks and 6 months compared to baseline [36]. All specific extraintestinal symptoms were reduced after 4 weeks, but some were unchanged after 6 months compared to baseline. The low-FODMAP group reported significantly lower diarrhea symptom scores compared to the SSRD group at 4 weeks (*p* = 0.024) and at 6 months (*p* = 0.023) (Appendix A).

### 2.4. Hormonal Changes

Apart from a weak difference in copeptin levels at baseline, there were no differences between the groups in the absolute hormone concentrations at any of the three time points (Table 2). Within the SSRD group, there was a significant decrease in leptin concentrations from baseline to 4 weeks (13.5 (6.8–22.0) µg/L vs. 10.4 (5.5–20.3) µg/L, *p* = 0.002) with an effect size of 0.37, indicating a small to moderate effect according to Cohen’s guidelines (Figure 1 and Table 2). At 6 months, the leptin concentrations in the SSRD group had returned to baseline levels (16.0 (6.5–25.5) µg/L, *p* = 0.670). No significant changes were observed in the copeptin, C-peptide, or insulin concentrations within the groups, but there was a tendency toward a significant decrease in the PAI-1 concentration at week 4 in the SSRD group (14.1 (8.7–21.6) ng/mL vs. 11.6 (6.9–19.3) ng/mL, *p* = 0.056), with an effect size of 0.23 (Table 2). When combining the groups, a significant decrease was observed from baseline to 4 weeks in both the leptin (*p* = 0.001) and PAI-1 (*p* = 0.019) concentrations. When calculating differences instead of absolute values, neither changes in the leptin concentration from baseline to 4 weeks nor changes in the PAI-1 concentrations differed significantly between groups (*p* = 0.103 and *p* = 0.531, respectively). The post hoc sensitivity analysis showed that the study was powered to detect a minimum effect size of 0.17 for within-group comparisons at 4 weeks and 0.23 at 6 months, as well as 0.23 and 0.28 for the corresponding between-group differences.

### 2.5. Comparison of Changes in Leptin and PAI-1 with Changes in Diet and Symptoms

The changed intake of kilocalories, carbohydrates, starch, and fat from baseline to 4 weeks was associated with changes in leptin concentrations (Table 3). When adjusted for weight, changes in leptin concentrations were significantly associated with the changes in fat intake (β: 0.057; 95% CI: 0.002–0.111; *p* = 0.041) and kilocalorie intake (β: 0.004; 95% CI: 0.000–0.008; *p* = 0.028) (Table 4). Changes in fat intake were also significantly associated with changes in leptin concentrations in the model including changes in weight and the intake of carbohydrates and protein (β: 0.071; 95% CI: 0.006–0.136; *p* = 0.032).

Regarding the association between changes in symptoms and leptin after 4 weeks, decreased leptin levels were associated with reduced diarrhea symptoms (β: 0.519; 95% CI: 0.048–0.990; *p* = 0.031). When changes in symptoms were analyzed in a model including changes in leptin levels, weight, and fat intake, no associations were found between changes in symptoms and either leptin or fat intake, but the decrease in weight was found to have a weak association with reduced belching or excess wind (β: 3.844; 95% CI: 0.132–7.556; *p* = 0.042) and urinary urgency (β: 3.530; 95% CI: 0.421–6.638; *p* = 0.026) (Table 5).

No associations were found between changes in PAI-1 and weight, nutrients, or symptoms, except for a weak association with belching/excess wind (*p* = 0.025) (Appendix A). Replacing weight with BMI did not influence the results or provide additional insights.

### 2.6. Principal Component Analysis

The first three independent components from the principal component analysis (PCA) together explained nearly 70% of the total variance. The first component, which explained 46% of the variance, correlated positively with changes in the intake of kilocalories, carbohydrates, disaccharides, sucrose, and added sugar. The second component correlated positively with changes in the intake of protein, fat, and fiber, whereas the third component correlated negatively with changes in monosaccharide intake and positively with changes in starch intake (Table 6).

The first principal component (captured by changes in the intake of kilocalories, carbohydrates, disaccharides, sucrose, and added sugar) showed a borderline significant association with changes in leptin levels (*p* = 0.060). Adjusting for other components did not affect the association, and it was only slightly reduced after adjusting for weight loss (*p* = 0.080). The association differed somewhat between the two groups, with a stronger association in the SSRD group compared to the low-FODMAP group, but this difference was not statistically significant based on the interaction effects (Table 7).

Changes in PAI-1 levels showed no significant associations with the first component, but the differences in association between the groups were also profound here, with a stronger association in the SSRD group compared to the low-FODMAP group, although the interaction effects were not significant (Table 7).

## 3. Discussion

The main findings of the present study were that the leptin and PAI-1 levels decreased after the 4-week dietary intervention in the entire study population. The reduction in leptin levels was significant in the SSRD group but not in the low-FODMAP group, whereas the PAI-1 levels decreased to a similar extent in both groups. An analysis of the nutrient intake and symptom changes within the overall study population revealed that a reduction in leptin levels was associated with decreases in the intake of kilocalories, carbohydrates, fat, and starch. However, no association was found between changes in leptin levels and symptom improvements. One component in the PCA that explained 46% of the variance correlated positively with changes in the intake of kilocalories, carbohydrates, disaccharides, sucrose, and added sugar. This component showed a borderline association with changes in leptin levels. The associations of both leptin and PAI-1 levels with this component were more pronounced in the SSRD group compared to the low-FODMAP group. Changes in weight and food intake during the study indicated good compliance with the diet. The hormonal changes returned to baseline levels at follow-up, when the participants had mainly returned to their ordinary food habits.

The most important findings of this dietary intervention were that both diets resulted in a significant overall reduction in GI and extraintestinal symptom scores [11]. However, no association was found between the reduction in any symptom and the reduction in leptin or PAI-1 levels. The only associations that remained after adjustments were weak associations between weight and a few specific extraintestinal symptoms, which is in alignment with an association between weight and extraintestinal symptoms [5]. As mentioned previously, the improvement in IBS symptoms following dietary changes may be due to interactions between luminal content, changes in gut endocrine cells mediating hormonal release, and weight loss [33,34]. Leptin is known to modulate several mechanisms by enhancing satiety signals and the release of other hormones, such as CCK and GLP-1 [29,30]. Through these hormones, not analyzed in the present study, leptin affects gut motility, nutrient digestion, and absorption [16]. However, this study did not identify any direct associations between leptin levels and IBS symptom improvements.

After the 4-week intervention, participants were free to eat whatever they preferred. As observed in the food diary books, most of the participants returned to dietary patterns that were similar to those prior to the study, although some changes remained, such as the decreased intake of certain carbohydrates. These long-term changes were most pronounced in the SSRD group. Consequently, the hormonal effects induced by the diet returned to the baseline levels at follow-up.

The current findings further strengthen the hypothesis that the SSRD diet decreases the leptin and PAI-1 concentrations in patients with IBS. In a previous study by Saidi et al. [31], a significant decrease was found in the leptin levels following a 4-week dietary intervention using the SSRD compared to the control group, consisting of IBS patients with no dietary intervention. In alignment with this, a diet with increased sugar and total kilocalories given to rats increased the leptin concentrations compared to the control group [37]. The former study found no association between decreased leptin concentrations and changes in carbohydrate intake when adjusting for weight [31], similarly to the current study. In a study by Roth et al. [32], a significant difference was found in changes in PAI-1 levels from baseline to 4 weeks between the SSRD group and the control group without any dietary intervention. In contrast, the previous findings of decreased concentrations of C-peptide and insulin after the SSRD could not be confirmed [31]. The use of non-fasting blood samples may have led to this discrepancy between the studies.

The adipose tissue is the largest endocrine organ in the body. Leptin levels have been found to be elevated in patients with IBS compared to healthy controls [38]. Interestingly, leptin levels are known to be directly proportional to body fat [23,26]. The bidirectional interaction between the adipose tissue and pancreatic beta cells, known as the adipo-insular axis, where adipokines and insulin are the main components, plays a pivotal role in metabolic regulation and the onset of metabolic disorders [39,40]. Insulin affects adipocyte metabolism and promotes leptin secretion, which inhibits insulin secretion under physiological conditions, acting as a dynamic feedback mechanism to regulate energy intake, storage, and expenditure [41,42]. However, in metabolic disorders such as obesity, the adipo-insular axis is dysregulated, with chronically increased leptin levels contributing to the release of pro-inflammatory cytokines, leptin resistance, insulin resistance, and beta cell dysfunction [39,41,43,44]. Furthermore, leptin may increase tight junction permeability [44], which may be of importance for the visceral hypersensitivity found in IBS [45].

At the 6-month follow-up, the leptin levels and weight had returned to the baseline levels. In contrast, the intake of various carbohydrates and the total IBS-SSS were still significantly lower at 6 months compared to baseline. However, since a large proportion of the participants included did not attend the 6-month follow-up, the sample size was small and there was a considerable risk of bias. The weak associations between leptin changes and nutrient changes may be due to other effects exerted on hormonal release. These effects are not only influenced by the absolute nutrient intake but also by other intermediate products and changes in the gut microbiota or short-chain fatty acid composition. Analyses of the gut microbiota composition and global metabolomics in relation to food intake and symptoms are needed to further explain the mechanisms underlying symptom relief. In addition, psychological influences may also affect the response rate regarding dietary interventions, since the gut–brain axis is considered important for the symptom experience in IBS [1,2,3].

The PAI-1 levels showed a tendency towards reduction in the SSRD group, with a significant change in the overall study population. Bariatric surgery resulting in weight loss leads to lower PAI-1 levels [46]. Several studies have confirmed that PAI-1 plays a central role in carcinogenesis and the development of metabolic diseases [18]. Thus, the reduction in the levels of PAI-1 appears to be important, and diets promoting weight loss and lower amounts of certain carbohydrates may be suitable as treatments for IBS and overweight.

Copeptin has been associated with an increased risk of developing diabetes mellitus [20]. In a meta-analysis on gestational diabetes mellitus, an association between copeptin and diabetes was, however, only found in the subgroup of overweight subjects [21]. No effect on copeptin levels was observed in the current study, in alignment with several studies that have shown no influence of nutrient intake on copeptin release [47,48], although more recent studies have demonstrated that fructose intake stimulates copeptin secretion [22].

The pathophysiology of IBS is complex and multifactorial, and the diagnosis is driven by the ROME IV criteria, without objective measurable signs. This study further highlights the difficulties in using hormones as reliable markers of IBS, due to their low specificity and correlations between hormones and other related illnesses, such as metabolic syndrome and diabetes [15,20,25]. Furthermore, the role of hormones in IBS is still unclear, and more research is needed to gain a deeper understanding of the underlying mechanisms. Nevertheless, the negative health effects associated with elevated circulating levels of leptin and PAI-1, as described above, underscore the importance of maintaining lower levels of these hormones. IBS is associated with overweight, metabolic syndrome, and hyperleptinemia [4,5,33,38], highlighting the need to reduce not only IBS symptoms but also those of other affected medical conditions. In the future, the SSRD should also be considered as a therapeutic option for other patient categories, such as those with overweight, metabolic syndrome, and diabetes, to reduce the negative health effects of these conditions, including chronically elevated hormone levels.

One of the main strengths of this study was the inclusion of two types of dietary interventions and the measurement of five hormones. By comparing two different diets, this study was able to identify which nutrients were more clearly associated with hormonal changes, thereby increasing the heterogeneity of the nutrient intake among the participants. Although the two diets have certain similarities, there are key differences to consider, such as the differences in carbohydrate and sugar intake between the groups. Another strength was the similar basal characteristics between the groups, making them more comparable. There are several limitations of this study. First and foremost, blood samples were collected from non-fasting participants, as this was chosen to ensure patients’ convenience in visiting the department in relation to their working hours. However, the participants usually visited the department at the same time at each follow-up, thus reducing the risk of irregularities between appointments. Another limitation was the relatively small cohort. The power analysis was performed to assess the response rate regarding symptom relief. However, a post hoc sensitivity analysis indicated that the study was powered to detect a minimum effect size between 0.17 and 0.28 for hormonal changes within and between the groups. Effects smaller than this threshold may not have been detectable with the available sample size, and it is possible that other hormonal changes would have been significant with a larger sample size. Additional hormones may be involved in gut function and dietary interventions and should be investigated in future studies.

In conclusion, the first hypothesis that hormone levels would change following a 4-week dietary intervention with either the SSRD or low-FODMAP diet was partly accepted, as two out of five hormones were affected. The 4-week dietary intervention with the SSRD led to lower levels of leptin in patients with IBS. The PAI-1 levels decreased in the overall study population. Certain dietary changes characteristic of the SSRD, such as reduced intake of carbohydrates, disaccharides, starch, sucrose, and added sugar, may affect leptin levels. All hormonal changes returned to baseline at follow-up, when the participants had mainly returned to their ordinary food habits. The second hypothesis—that changes in hormone concentrations would be associated with changes in symptoms—was rejected. Improvements in symptoms may instead have been caused by dietary and weight changes, rather than by hormonal changes. Further studies are warranted with extended psychological and molecular analyses to better understand the broader therapeutic context of nutritional interventions.

## 4. Materials and Methods

### 4.1. Study Design

This study was an open, randomized, non-inferiority trial with two parallel groups, conducted from March 2022 to February 2024 at the Department of Internal Medicine, Skåne University Hospital in Malmö, Sweden [49]. Following a 10-day run-in period (baseline), the participants were randomized to undergo a 4-week dietary intervention with either the SSRD or a low-FODMAP diet. Following the intervention, participants received information about the alternative diet to which they were not initially randomized and were allowed to try it voluntarily without any obligation to adhere to specific dietary restrictions. However, participants in the low-FODMAP group were required to reintroduce FODMAP-containing foods gradually, one by one, according to clinical guidelines, prior to the 6-month follow-up [7,8]. Data collection occurred at baseline and after 4 weeks and 6 months, during which the participants completed a set of assessments, including the study questionnaire, a food diary [50], the ROME IV questionnaire [50], the IBS-SSS [36], and the Visual Analog Scale for Irritable Bowel Syndrome (VAS-IBS) [35]. Furthermore, participants underwent physical examinations, including abdominal palpation and measurements of their body weight, height, and waist circumference. Blood samples were collected for later hormone analyses at the Departments of Clinical Chemistry at Gothenburg or Malmö [51,52,53] (Figure 1).

### 4.2. Participants

The inclusion criteria for this study included a diagnosis of IBS according to the ROME IV criteria [1], independently of the IBS subtype; an age range of 18 to 70 years; and an IBS-SSS score greater than 175. Exclusion criteria included the following: alcohol and/or drug abuse, current eating disorders, pregnancy, the presence of any organic GI disease, a history of major GI surgery, severe organic or psychiatric diseases, severe food allergies, or already following gluten-free, vegan, low-FODMAP, or low-carbohydrate high-fat (LCHF) diets.

A comprehensive description of the recruitment process has been previously provided [49]. In summary, a data search was conducted using medical records in the County of Region Skane using the International Classification of Diseases (ICD) Revision 10 to identify patients diagnosed with one of the following subtypes of IBS between 2019 and 2022: K58.1 (IBS-D), K58.2 (IBS-C), K58.3 (IBS-M), and K58.8 (IBS-U). A total of 744 patients were contacted by letter and phone call. To enhance recruitment, information leaflets were sent to 203 primary healthcare centers in the county, and several lectures were held for healthcare staff. Furthermore, a professional recruitment agency (Trialy, Gothenburg, Sweden) was engaged to utilize social media platforms to recruit patients with a confirmed IBS diagnosis. A total of 300 eligible patients were identified, of whom 214 were randomized to either the SSRD or low-FODMAP group through block randomization. Following the initial screening, 155 patients (72.4% of randomized cases) proceeded to the dietary intervention, with exclusions occurring due to failure to attend the first visit or not meeting the inclusion or exclusion criteria. Patients recruited through social media had an inclusion rate of 42.7%, whereas those identified through medical records had an inclusion rate of 6.5%. Seven of the participants included had a total IBS-SSS slightly lower than 175 but were included due to an evident diagnosis of IBS. Furthermore, 13 participants were excluded since no blood samples were collected at two separate time points, either due to dropouts or due to missing samples. Thus, 142 participants were included in the final analysis (Figure 2).

### 4.3. Dietary Advice

Participants received both verbal and written information about their randomized diets. For the SSRD group, the dietary intervention focused on reducing starch and sucrose intake and increasing the intake of certain fruits, vegetables, fish, meat, and dairy products. The dietary advice was modified from the guidelines developed for patients with congenital sucrase-isomaltase deficiency (CSID) [55,56] and have previously been described in detail [57]. In short, all sucrose-containing foods had to be avoided. Participants were allowed one serving per day of either whole grain bread or oatmeal porridge. Participants were recommended whole grains instead of processed cereals for breakfast. Fiber-rich alternatives to pasta and rice were recommended. Smaller amounts of regular white rice and pasta were allowed for participants who did not tolerate a high-fiber diet. Intake of unprocessed pork, beef, lamb, turkey, chicken, and eggs was allowed. Participants were recommended to avoid processed meat if it contained sugar or starch. Intake of natural dairy products was allowed, but not oat or soy milk. The use of salt, pepper, and fresh herbs was unrestricted. Nuts were recommended as alternatives to sugary snacks. Participants were encouraged to increase their protein and/or fat intake and prolong chewing to increase the salivary amylase breakdown of starch and delay GI transport. Participants received a list of fruits and vegetables containing less starch (Appendix A) [57].

For the low-FODMAP group, the dietary intervention focused on reducing the intake of fructans (e.g., wheat and wheat-containing foods such as pasta, asparagus, broccoli, cabbage, garlic, and onion), galacto-oligosaccharides (e.g., pulses such as beans, lentils, peas, and chickpeas), lactose (e.g., milk and other dairy products), fructose more than glucose (e.g., honey, apples, melon, and pears), and polyols (e.g., apples, peaches, pears, plums, raspberries, and strawberries). Following the 4-week intervention, participants were required to reintroduce FODMAP-containing foods, according to clinical routines, since the low-FODMAP diet excludes several food items [54,58]. The greatest differences between the two diets are that the SSRD aims to reduce starch, sucrose, and added sugar intake but allows the intake of fructose, lactose, and sweeteners [56], whereas the low-FODMAP diet is less restrictive regarding starch and sucrose [8,54].

Both groups received recipes for meals suitable for their respective diets to facilitate compliance. All participants were instructed to maintain their caloric intake, physical activity levels, ongoing medications, probiotics, and supplements throughout the study. Participants received no advice regarding food intake frequency or regularity. Furthermore, participants were instructed not to make any changes to or introduce new medications or diets during the 4-week intervention.

### 4.4. Questionnaires

#### 4.4.1. Study Questionnaire

A questionnaire regarding sociodemographic factors, lifestyle habits, pregnancy and childbirth, medical history, drug treatments, and family history was completed by all participants. Food intake was digitally recorded using the Riksmaten Flex 2021 platform developed by the Swedish Food Agency [50]. Participants were instructed to record their food intake over 3 days (Wednesday to Friday) at baseline, after 4 weeks, and at the 6-month follow-up. The SCOFF (Sick, Control, One, Fat, Food) questionnaire was used to screen for symptoms of eating disorders at baseline, after 4 weeks, and at 6 months [59].

#### 4.4.2. ROME IV Questionnaire

Questions 40–48 were used in the Swedish version of the ROME IV questionnaire to diagnose IBS. A license for the use of this tool was obtained from the ROME Foundation, Inc., Raleigh, NC, USA.

The ROME IV questionnaire has sensitivity of 62.7% and specificity of 94.5% in diagnosing IBS [60]. For FBD, the sensitivity is 54.7% and the specificity is 93.3% [60].

#### 4.4.3. Irritable Bowel Syndrome Severity Scoring System

The IBS-SSS estimates abdominal pain, abdominal distension, satisfaction with bowel habits, and the impact of bowel habits on daily life using a visual analog scale (VAS) ranging from absent (0 mm) to very severe (100 mm), along with a measure of the number of days with abdominal pain over the past 10 days [36]. The highest achievable score is 500 for the total IBS-SSS score. Scores between 75 and 174 indicate a mild symptom burden, 175–299 indicate a moderate symptom burden, and ≥300 indicate a severe symptom burden. Extraintestinal symptoms, including nausea, difficulties eating a whole meal, headache, back pain, belching/excess wind, reflux, urinary urgency, leg pain, muscle/joint pain, and fatigue, were estimated on a VAS scale with a maximum achievable score of 500 [36].

#### 4.4.4. Visual Analog Scale for Irritable Bowel Syndrome

The VAS-IBS questionnaire was used to estimate abdominal pain, diarrhea, constipation, bloating and flatulence, vomiting and nausea, intestinal symptoms’ influence on daily life, and psychological well-being. These symptoms were measured using VAS scales ranging from absent (0 mm) to very severe (100 mm). In this study, the VAS scales were inverted compared to the original version, and reference values for healthy individuals have been previously described [35,61].

### 4.5. Blood Samples

Blood samples were collected and analyzed at baseline, after 4 weeks, and at the 6-month follow-up. Blood samples were drawn into ethylenediaminetetraacetic acid (EDTA) tubes (Sigma-Aldrich, Merck KGaA, Darmstadt, Germany) and BD vacutainer serum separator tubes (SST) (KRUUSE, Langeskov, Denmark). Plasma was centrifugated for 15 min at 2000× *g* and frozen at −70 °C. Serum was kept at room temperature for 30 min until coagulation and thereafter centrifugated for 10 min at 2000× *g* and frozen at −70 °C. Samples for the analysis of C-peptide and insulin were centrifugated again after thawing before analysis. Participants were non-fasting prior to testing. Serum levels of copeptin and leptin were analyzed at the Department of Clinical Chemistry at Sahlgrenska University Hospital in Gothenburg, Sweden, by enzyme-linked immunosorbent assay (ELISA) and chemiluminescence, respectively [51,52]. The plasma levels of PAI-1 and serum levels of C-peptide and insulin were analyzed according to routine analysis at the Department of Clinical Chemistry at Skåne University Hospital in Malmö, Sweden [52]. An immunochemical assay called Chromolize PAI-1 was used to analyze PAI-1, and the immunometric sandwich method with the electrochemiluminescence immunoassay (ECLI) detection technique based on a ruthenium derivative was used to analyze C-peptide and insulin [53]. One participant in the low-FODMAP group had missing blood samples at baseline. Coefficients of variance (CV) for the different analyses are given in Appendix A.

### 4.6. Statistical Analysis

Power calculation was based on non-inferiority, where the SSRD was tested against a standard treatment (low FODMAP). The primary outcome was the responder rate (RR = ∆Total IBS-SSS ≥ −50) and was assumed to be 65% in both treatment groups at week 4. A difference in the responder rate as large as 20% in favor of the standard treatment would allow the SSRD to be non-inferior. The sample size based on 80% power, a one-sided confidence level of 97.5%, and an expected 10% loss at follow-up to confirm non-inferiority was calculated to be 100 patients in each group. Due to a few dropouts, the study was closed after the inclusion of 155 patients, following a second consultation with the statistician. A post hoc sensitivity analysis was conducted to calculate the minimum detectable effect sizes (MDESs) for the hormonal changes. The MDES was defined as the smallest Wilcoxon effect size (r) that the study was powered to detect with 80% power and a significance level of α = 0.05. The minimum detectable effect size (MDES) was estimated using a paired *t*-test for within-group comparisons, assuming a correlation of 0.5 between paired observations, and an independent-samples *t*-test was used for between-group comparisons. The smallest detectable effect size was then standardized to Cohen’s d and converted to a Wilcoxon effect size (r) using the approximation r = d/√(d^2^ + 4).

Statistical analyses were performed using IBM SPSS, version 29. The Kolmogorov–Smirnov test was used to determine the variable distribution. Age, kilocalorie intake, and certain macronutrients were normally distributed and were therefore presented as the mean and ± standard deviation (SD), and data were calculated via a paired-samples *t*-test for comparisons within the groups regarding changes from baseline to the end of the 4-week intervention and to follow-up and an independent-sample *t*-test for comparisons between the groups. All other variables were not normally distributed and were presented as medians (interquartile ranges), using Wilcoxon signed ranks for comparison within the groups and the Mann–Whitney U-test for comparisons between groups. Effect sizes (r) were calculated using the standardized Wilcoxon effect size (r), defined as the absolute value of the test statistic Z divided by the square root of the number of observations (r = Z/√N). Values of r were interpreted as small (0.2), medium (0.5), or large (0.8) effects, according to Cohen’s guidelines. Fisher’s exact test was used for categorical variables, which were presented as numbers and percentages. Spearman’s correlation test was used for correlation analyses. Missing values were excluded from the analysis. Since the leptin and PAI-1 concentrations changed during the intervention, associations between changes in these hormones, nutrients, and symptoms were analyzed using the generalized linear model. First, changes in leptin and PAI-1 levels were used as the dependent variable to calculate associations with weight and nutrient intake (predictors). Second, the analyses with nutrients were repeated after being adjusted for weight, since weight is established as a regulator of leptin and PAI-1 levels [25,26,46]. Third, changes in hormone levels were analyzed in a model adjusting simultaneously for changes in weight and carbohydrate, protein, and fat intake. All analyses were repeated with weight replaced by BMI. Since the results were similar for both parameters and did not influence the outcome, weight was utilized in the main analysis for clarity, although either parameter could have been reported. Finally, changes in symptom scores were used as dependent variables, with changes in leptin or PAI-1 levels as predictors in an unadjusted model. Changes in symptom scores were then further analyzed in a model including changes in leptin levels, fat intake, and weight as independent variables. These variables were chosen because fat intake remained associated with changes in leptin after adjustment for weight, and weight (correlated with kilocalorie intake) has been found to be associated with leptin levels and symptoms [5,25]. β coefficients and 95% confidence intervals (CIs) are given.

PCA was used to identify patterns and reduce the dimensionality of changes in nutrient intake following the dietary intervention. To perform the PCA, nutrient variables were first standardized, after which eigenvalues and eigenvectors were computed to obtain the principal components. Components with a proportion of variance explained of more than 10% were retained and further used in linear regression models of logarithmic values to examine the associations with changes in leptin and PAI-1 levels. Differences in association between the groups were used by adding an interaction effect in the regression models. For all statistical tests, *p* < 0.05 was considered statistically significant.

## Figures and Tables

**Figure 1 ijms-26-04607-f001:**
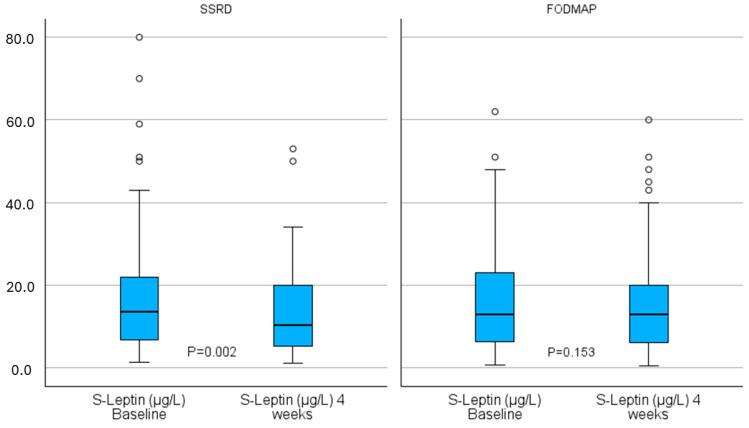
Leptin concentrations at baseline and 4 weeks, stratified by group. SSRD = starch- and sucrose-reduced diet. Low FODMAP = low content of fermentable oligo-, di-, and monosaccharides and polyols. Boxplot of leptin concentrations at baseline and 4 weeks. Wilcoxon signed ranks. *p* < 0.05 was considered statistically significant.

**Figure 2 ijms-26-04607-f002:**
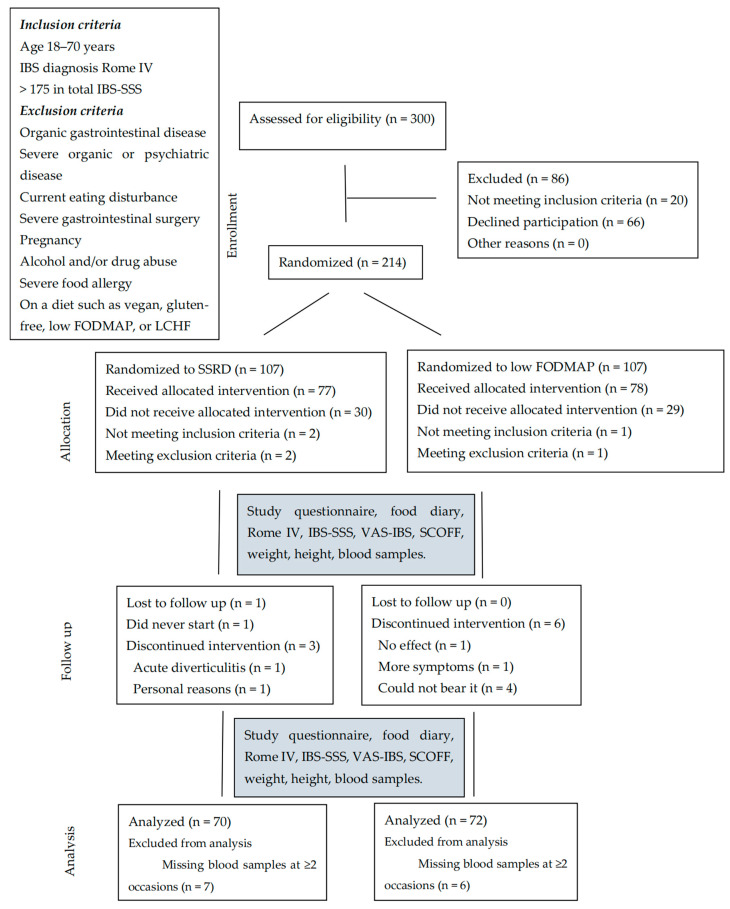
Flow chart. Study inclusion and exclusion criteria. SSRD = starch- and sucrose-reduced diet. Low FODMAP = low content of fermentable oligo-, di-, and monosaccharides and polyols. IBS-SSS = irritable bowel syndrome severity scoring system [36]. VAS-IBS = Visual Analog Scale for IBS [35]. LCHF = low-carbohydrate high-fat. SCOFF = Sick, Control, One, Fat, Food [54].

**Table 1 ijms-26-04607-t001:** Basal characteristics.

Parameter	SSRD N = 70	Low FODMAP N = 72	*p*-Value
**Sex (women/men; n, %)**	57 (81)/13 (19)	64 (89)/8 (11)	0.243
**Age (year)**	42.4 ± 14.0	45.2 ± 14.2	0.234
**Disease duration (year)** Missing	19.5 (7.8–28.8)	20.0 (10.5–33.5) 3	0.313
**Education (n, %)**			0.578
Primary school	5 (7.1)	2 (2.8)	
Secondary school	10 (14.3)	13 (18.1)	
Education after secondary school	18 (25.7)	16 (22.2)	
Examination at university	37 (52.9)	41 (56.9)	
**Occupation (n, %)**			0.684
Working full-time	44 (62.9)	45 (62.5)	
Working 99–51%	6 (8.6)	8 (11.1)	
Working 50%	1 (1.4)	1 (1.4)	
Studying	8 (11.4)	9 (12.5)	
Sick leave	3 (4.3)	0 (0)	
Unemployment	2 (2.9)	1 (1.4)	
Retirement	6 (8.6)	8 (11.1)	
**Marital status (n, %)**			0.595
Living alone	15 (21.4)	12 (16.7)	
Living together	50 (71.4)	52 (72.2)	
Other	5 (7.1)	8 (11.1)	
**Smoking (n, %)**			0.131
Never	38 (54.3)	39 (54.2)	
Former	24 (34.3)	26 (36.1)	
Present irregular	2 (2.9)	6 (8.3)	
Present regular	6 (8.6)	1 (1.4)	
**Alcohol intake for 1 week (standard glass) (n, %)**			0.882
Missing	-	1	
<1	30 (42.9)	29 (40.8)	
1–4	26 (37.1)	29 (40.8)	
5–9	12 (17.1)	12 (16.9)	
10–14	1 (1.4)	1 (1.4)	
≥15	1 (1.4)	0 (0)	
**Physical activity for 1 week (n, %)**			0.274
Missing	-	1	
No time	10 (14.3)	5 (7.0)	
<30 min	10 (14.3)	12 (16.9)	
30–60 min	13 (18.6)	14 (19.7)	
60–90 min	7 (10.0)	16 (22.5)	
90–120 min	8 (11.4)	8 (11.3)	
>120 min	22 (31.4)	16 (22.5)	

SSRD = starch- and sucrose-reduced diet. Low FODMAP = low content of fermentable oligo-, di-, and monosaccharides and polyols. Values are given as numbers and percentages, mean ± standard deviation, or median (interquartile range). Physical activity means activity that causes shortness of breath. Fisher’s exact test, independent-sample *t*-test (age), and Mann–Whitney U test. *p* < 0.05 was considered statistically significant.

**Table 2 ijms-26-04607-t002:** Changes in hormone levels and differences between the groups.

	SSRD N = 70			Low FODMAP N = 72			(§) *p*-Value	Effect Size
Variable	Median (IQR)	*p*-Value	Effect Size	Median (IQR)	*p*-Value	Effect Size		
**S-Copeptin (pmol/L)**								
Baseline Missing	4.0 (2.8–6.3)	-		3.0 (2.0–4.0) 1	-		**0.044**	0.17
**4 weeks**	4.0 (3.0–6.0)	0.466	0.07	3.0 (2.0–4.0)	0.802	−0.02	0.101	0.14
6 months Missing	4.0 (2.0–5.0) 21	0.081	0.25	3.0 (2.0–4.0) 24	0.893	−0.02	0.118	0.16
**S-Leptin (µg/L)**								
Baseline Missing	13.5 (6.8–22.0)	-		13.0 (6.2–23.0) 1	-		0.790	0.02
**4 weeks**	10.4 (5.1–20.3)	**0.002**	0.37	13.0 (6.2–20.0)	0.153	0.18	0.349	−0.08
6 months Missing	16.0 (6.5–25.5) 21	0.670	0.06	16.0 (7.6–23.5) 24	0.917	0.01	0.848	0.02
**P-PAI-1 (ng/mL)**								
Baseline Missing	14.1 (8.7–21.6)	-		13.7 (7.4–26.6) 1	-		0.910	0.01
4 weeks	11.6 (6.9–19.3)	0.056	0.23	12.5 (5.6–21.5)	0.171	0.16	0.845	−0.02
6 months ***Missing***	9.9 (6.3–19.2) 21	0.551	0.08	9.9 (7.7–21.2) 24	0.262	0.16	0.823	−0.02
**P-C-Peptide (nmol/L)**								
Baseline Missing	1.0 (0.8–1.5)	-		1.1 (0.7–1.5) 1	-		0.867	−0.01
**4 weeks**	1.0 (0.8–1.3)	0.237	0.14	1.1 (0.8–1.4)	0.551	−0.08	0.070	−0.15
6 months Missing	1.1 (0.7–1.4) 21	0.970	−0.006	1.0 (0.7–1.4) 24	0.788	−0.04	0.621	0.05
**P-Insulin (mIE/L)**								
Baseline Missing	13.0 (7.8–26.0)	-		14.0 (7.0–26.0) 1	-		0.845	−0.02
**4 weeks**	13.0 (8.0–20.0)	0.210	0.14	13.0 (8.0–23.3)	0.701	−0.05	0.451	−0.06
6 months Missing	15.0 (7.0–28.5) 21	0.795	−0.04	11.0 (6.3–28.0) 24	0.436	−0.11	0.453	0.08

SSRD = starch- and sucrose-reduced diet. Low FODMAP = low content of fermentable oligo-, di-, and monosaccharides and polyols. PAI-1 = plasminogen activator inhibitor-1. S = serum. P = plasma. One blood sample was missing at baseline in the low-FODMAP group. Serum concentration values are given as medians (interquartile range (IQR)). Wilcoxon signed ranks for comparisons within the groups and Mann–Whitney U test (§) for comparisons between the two groups. *p* < 0.05 was considered statistically significant and marked in bold. The Wilcoxon effect size (r) was calculated as Z/√N, where Z is the test statistic from the Wilcoxon signed-rank test for comparison within the groups and the test statistic from the Mann–Whitney U test for comparison between the two groups. N is the number of observations.

**Table 3 ijms-26-04607-t003:** Association of changes in leptin (dependent variable) with changes in weight and nutrient intake.

Independent Variable	β-Coefficient and 95% CI	*p*-Value
Weight (kg)	1.036 (−0.325–2.398)	0.136
Kilocalorie intake (kcal)	0.004 (0.001–0.008)	**0.020**
Carbohydrate intake (g)	0.033 (0.003–0.064)	**0.033**
Protein intake (g)	−0.007 (−0.091–0.078)	0.879
Fat intake (g)	0.056 (0.001–0.110)	**0.044**
Fiber intake (g)	0.177 (−0.119–0.474)	0.241
Monosaccharide intake (g)	0.117 (−0.026–0.260)	0.109
Disaccharide intake (g)	0.023 (−0.050–0.097)	0.536
Sucrose intake (g)	0.042 (−0.019–0.012)	0.357
Starch intake (g)	0.064 (0.004–0.124)	**0.037**
Added sugar intake (g)	0.030 (−0.039–0.099)	0.392

Generalized linear model for calculation of association between changes in leptin concentration from baseline to 4 weeks as dependent variable and changes in weight and nutrient intake as independent variables. Values are presented as β-coefficients and 95% confidence intervals (CIs). *p* < 0.05 was considered statistically significant and marked in bold.

**Table 4 ijms-26-04607-t004:** Association of changes in leptin (dependent variable) with changes in nutrient intake, adjusted for changes in weight.

Adjusted Independent Variable	β-Coefficient and 95% CI	*p*-Value
Kilocalorie intake (kcal)	0.004 (0.000–0.008)	**0.028**
Carbohydrate intake (g)	0.031 (−0.001–0.062)	0.059
Fat intake changes (g)	0.057 (0.002–0.111)	**0.041**
Fiber intake (g)	0.179 (−0.116–0.473)	0.235
Monosaccharide intake (g)	0.128 (−0.015–0.271)	0.080
Disaccharide intake (g)	0.018 (−0.056–0.092)	0.630
Sucrose intake (g)	0.036 (−0.054–0.126)	0.433
Starch intake (g)	0.059 (−0.002–0.120)	0.058
Added sugar intake (g)	0.023 (−0.047–0.093)	0.517

Generalized linear model for calculation of association between changes in leptin concentration from baseline to 4 weeks as dependent variable and changes in nutrient intake as independent variables, adjusted for weight changes. Values are presented as β-coefficients and 95% confidence intervals (CIs). *p* < 0.05 was considered statistically significant and marked in bold.

**Table 5 ijms-26-04607-t005:** Association of changes in symptoms with changes in leptin, weight, and fat.

Symptom Change (Dependent Variable)	Leptin Change β-Coefficient and 95% CI; *p*-Value	Weight Change β-Coefficient and 95% CI; *p*-Value	Fat Intake Change β-Coefficient and 95% CI; *p*-Value
Abdominal pain	0.098 (−0.288–0.485); 0.619	−0.774 (−3.926–2.378); 0.630	0.035 (−0.090–0.160); 0.583
Diarrhea	0.456 (−0.035–0.948); 0.069	−2.186 (−6.193–1.822); 0.285	−0.002 (−0.161–0.157); 0.980
Constipation	−0.105 (−0.547–0.337); 0.642	−0.761 (−4.366–2.845); 0.679	−0.062 (−0.206–0.081); 0.392
Bloating and flatulence	−0.007 (−0.446–0.432); 0.975	1.337 (−2.236–4.909); 0.463	−0.011 (−0.154–1.33); 0.883
Vomiting and nausea	−0.234 (−0.540–0.072); 0.134	−0.382 (−2.873–2.108); 0.764	0.004 (−0.095–0.103); 0.937
Intestinal symptoms’ influence on daily life	0.033 (−0.381–0.447); 0.877	1.354 (−2.023–4.731); 0.432	0.030 (−0.104–0.164); 0.663)
Psychological well-being	0.061 (−0.310–0.431); 0.749	−1.040 (−4.064–1.985); 0.500	−0.108 (−0.228–0.012); 0.077
Difficulties eating a whole meal	−0.118 (−0.413–0.178); 0.434	−1.624 (−4.032–0.784); 0.186	0.027 (−0.068–0.123); 0.578
Headache	−0.108 (−0.463–0.248); 0.553	2.045 (−0.854–4.943); 0.167	0.038 (−0.077–0.153); 0.518
Back pain	0.038 (−0.322–0.398); 0.837	1.807 (−1.127–4.742); 0.227	0.066 (−0.051–0.182); 0.270
Fatigue	0.115 (−0.246–0.477); 0.532	1.484 (−1.463–4.432); 0.324	−0.025 (−0.142–0.092); 0.676
Belching/excess wind	−0.084 (−0.539–0.371); 0.718	3.844 (0.132–7.556); **0.042**	0.042 (−0.105–0.190); 0.574
Reflux	−0.123 (−0.503–0.258); 0.527	2.464 (−0.637–5.565); 0.119	−0.012 (−0.135–0.111); 0.848
Urinary urgency	−0.324 (−0.705–0.057); 0.096	3.530 (0.421–6.638); **0.026**	0.012 (−0.111–0.136); 0.845
Leg pain	−0.167 (−0.421–0.086); 0.195	1.254 (−0.813–3.320); 0.234	−0.001 (−0.083–0.081); 0.982
Muscle/joint pain	−0.008 (−0.367–0.351); 0.964	−1.787 (−4.713–1.139); 0.231	0.041 (−0.075–0.157); 0.486
Total IBS-SSS	0.154 (−1.346–1.653); 0.841	0.223 (−12.007–12.453); 0.971	0.336 (−0.150–0.821); 0.175
Total extraintestinal IBS-SSS	−0.510 (−1.462–0.443); 0.294	6.335 (−1.433–14.104); 0.110	0.097 (−0.212–0.405); 0.538

Symptoms measured by irritable bowel syndrome severity scoring system (IBS-SSS) [36] and visual analog scale for IBS (VAS-IBS) [35]. Generalized linear model for calculation of association between changes in symptom scores (dependent variables) from baseline to 4 weeks and changes in leptin, weight, and fat intake as independent variables. Values are presented as β-coefficients and 95% confidence intervals (CIs). *p* < 0.05 was considered statistically significant and marked in bold.

**Table 6 ijms-26-04607-t006:** Factor loadings of three components based on principal component analysis.

Nutrient Intake Change	Component 1	Component 2	Component 3
Kilocalories	**0.40**	0.30	−0.01
Carbohydrates	**0.42**	−0.10	0.19
Protein	0.18	**0.48**	−0.05
Fat	0.26	**0.42**	−0.18
Fiber	0.22	**0.42**	0.07
Monosaccharides	0.19	0.03	**−0.44**
Disaccharide	**0.40**	−0.29	−0.08
Sucrose	**0.39**	−0.33	−0.06
Starch	0.16	0.05	**0.85**
Added sugar	**0.38**	−0.35	−0.06
Proportion of variance explained	46%	21%	11%

The first three independent components from the PCA analysis explained together nearly 70% of the total variance. The first component correlated positively with changes in the intake of kilocalories, carbohydrates, disaccharides, sucrose, and added sugar. The second component correlated positively with changes in protein, fat, and fiber. The third component correlated negatively with changes in monosaccharide intake and positively with changes in starch intake. Correlations marked in bold.

**Table 7 ijms-26-04607-t007:** Associations between the first component and changes in leptin and PAI-1 levels.

Change	All N = 133	SSRD N = 66	Low FODMAP N = 67	Difference in Association Between Groups
**Changes in ln S-Leptin (** **β; p-value)**				
Component 1	0.89 (0.06)	1.04 (0.17)	0.26 (0.68)	−0.78 (0.44)
Adjusted for component 2	0.88 (0.06)	1.02 (0.18)	0.11 (0.87)	−0.93 (0.35)
Adjusted for component 3	0.88 (0.05)	1.15 (0.14)	0.27 (0.68)	−0.83 (0.41)
Adjusted for change in weight	0.81 (0.08)	0.99 (0.19)	0.32 (0.61)	−0.79 (0.43)
**Changes in ln P-PAI-1 (** **β; p-value)**				
Component 1	0.85 (0.43)	2.12 (0.12)	−0.35 (0.86)	−2.46 (0.29)
Adjusted for component 2	0.86 (0.43)	2.03 (0.12)	−0.38 (0.85)	−2.80 (0.23)
Adjusted for component 3	0.85 (0.44)	2.27 (0.10)	−0.42 (0.83)	−2.37 (0.31)
Adjusted for change in weight	0.82 (0.46)	2.08 (0.13)	−0.33 (0.87)	−2.47 (0.29)

ln = logarithmic value. PAI-1 = plasminogen activator inhibitor-1. S = serum. P = plasma. Associations with the first component were calculated for the differences in hormonal levels between baseline and 4 weeks. Differences in association between groups calculated as interaction effects. Linear regression models. Values are given as β coefficients. *p* < 0.05 was considered statistically significant.

## Data Availability

The data presented in this study are available on request from the corresponding author due to ethical reasons.

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
