# Peer review of "Leptin and PAI-1 Levels Are Decreased After a Dietary Intervention in Patients with Irritable Bowel Syndrome"

_ijms, 2025, doi:10.3390/ijms26104607_

Round 1
Reviewer 1 Report
Comments and Suggestions for Authors
This paper studies the effect of 2 diets on IBS patients, regarding symptoms, and hormones copeptin, leptin, PAI-1, C-peptide, and insulin. Patients followed a 4 week intervention diet and were followed up for 6 months.
The trials and the statictics are well designed and applied, and IBS is a less explored condition that should be given attention. The paper is alltogether novel and brings interesting results regarding leptin, although as the authors point out: effects are reversible.
The paper is difficult to read, long tables are hard to interpret and labels are confusing. Authors should show results in easier ways, levels of significance should be added for easier interpreting. The text should be made more concise.
I believe with some thorough editing it can be improved.
Reviewer 2 Report
Comments and Suggestions for Authors
The study investigates the effects of two dietary interventions—starch and sucrose reduced diet (SSRD) and low FODMAP diet—on hormone levels and symptom relief in patients with irritable bowel syndrome (IBS). the results achieved are encouraging because showed that SSRD significantly reduced leptin levels, while no significant change occurred in the low FODMAP group. Therefore of good quality and constant and analytical work that does not justify the journal to which it was submitted. I recommend a specialized journal of nutrition or diagnostics.
Reviewer 3 Report
Comments and Suggestions for Authors
This manuscript presents a well-designed randomized controlled study comparing the effects of two dietary interventions—SSRD and low-FODMAP—on hormonal markers in individuals with IBS. The research question is relevant and timely, and the findings contribute to the understanding of how specific diets influence biomarkers like leptin and PAI-1 in this population. The study is methodologically sound overall, with a thoughtful application of statistical tools, and it provides intriguing insights into diet-hormone interactions. However, to enhance the clarity, reproducibility, and scientific rigor of the work, several areas should be improved:
- It's essential to clearly define the inclusion criteria, such as age range, IBS diagnosis, and specific IBS subtypes (IBS-D, IBS-C, or IBS-M). Additionally, outlining exclusion criteria—such as comorbidities, medications that might interfere with hormonal levels, or prior treatments—will enhance transparency and aid readers in assessing the study's generalizability.
- A comprehensive breakdown of the SSRD and low-FODMAP diets is necessary. This should include:
-
- A list of specific foods or food groups allowed or restricted in each diet.
- Information on monitoring dietary adherence (e.g., food diaries, participant check-ins with dietitians).
- Any caloric or macronutrient adjustments made during the intervention should also be detailed.
3. Specify the assay types used (e.g., ELISA, immunoassays) and detail the manufacturers or protocols followed. Important aspects to cover include:
-
- Assay validation, intra-assay variability, and detection limits for each hormone.
- Sample processing details, such as timing of blood draws and storage conditions.
4. It's essential to provide a clearer description of how missing data were managed (e.g., imputation methods or exclusion criteria for missing values). Furthermore:
-
- Clarify the rationale for choosing specific statistical models, like generalized linear models (GLM) for hormone changes and symptom scores.
- Elaborate on the assumptions of the statistical tests (e.g., normality and homogeneity of variance for parametric tests) and how these were validated before analysis.
- Consider including post-hoc power analysis or sample size calculations to verify sufficient power to detect meaningful differences.
5. The conclusion should emphasize the lack of correlation between hormonal markers and symptom improvement, a key finding that adds complexity to diet-related interventions. Revising the conclusion to reflect why hormonal markers such as leptin and PAI-1 may not correlate directly with IBS symptoms could be important, indicating that other biomarkers or mechanisms might be more pertinent for symptom relief.
6. Discuss potential research directions, like examining other hormonal markers that may correlate more closely with IBS symptoms or studying the role of gut microbiota in influencing diet and hormonal levels.
7. Offer recommendations for clinicians, stressing that dietary modifications alone might not suffice for symptom relief and highlighting the need for further studies to grasp the broader therapeutic context of nutritional interventions.
7. Provide a more precise rationale for selecting specific statistical models for symptom data, addressing why linear regression versus generalized linear models was chosen. Also, justify the use of BMI versus weight as covariates in the models.
8. Including effect sizes (e.g., Cohen's d, partial eta-squared) would help readers understand the magnitude of the observed differences, particularly regarding hormonal changes.
9. Conducting and reporting a post-hoc power analysis can verify that the sample size was adequate for detecting significant differences in key outcomes (hormonal changes and symptoms). If a power analysis wasn't performed beforehand, provide justification based on prior research or expected effect sizes.
Comments on the Quality of English LanguageThe English quality of the manuscript is adequate but can be improved for better clarity and readability. While the scientific content is explicit, language polishing is needed to correct awkward phrasing, improve sentence structure, and ensure consistency in tense and terminology. Key issues to address include:
1. Grammatical errors and vague expressions.
2. Repetitive phrases that can be simplified.
3. Inconsistent use of scientific and statistical terminology.
A review by a native English speaker or professional editor is recommended to meet international publication standards.
Reviewer 4 Report
Comments and Suggestions for Authors
Dear editor
I read with interest the article: “Leptin and PAI-1 levels are decreased after a dietary intervention in patients with irritable bowel syndrome” sent to your journal.
This study is aimed to investigate the hormonal status of patients with IBS who underwent either a low-FODMAP diet or a sucrose reduced diet (SSRD). The authors enrolled a total of 142 patients and assessed the levels of 5 hormones other than IBS’ symptoms. It was found that both diets reduced gastrointestinal and extraintestinal symptoms after 4 weeks, but only SSRD significantly decreased leptin levels. PAI-1 levels decreased in both groups. Hormonal changes were linked to dietary intake but did not correlate with symptom improvements.
The article is well written, the methodology is clearly explained and the statistical analysis is comprehensible.
However, some comments are needed:
The authors followed an adequate selection and randomization mechanism; however, the authors should describe whether patients had baseline gastroenterologically active therapy. Did any of the patients modify their therapy during the 8-week follow-up?
Why was the dietary recall performed on a 3 days recall questionnaire and not a daily one? This induces a significant recall bias.
Authors should provide the list of food items advised in both diets.
Can the Authors justify why no comparison group was evaluated as well? Both the SSRD and low‑FODMAP groups lost weight and calories, however any diet that reduces energy intake will lower leptin and other adipocyte-related hormone; A hypocaloric diet control would better show whether the improvement in aabdominal and extra-abdominal symptom is related to the diet itself or the hormone change.
The decision to not perform the hormone assessment while fasting induces a very important bias that limits the results. Similarly, was the time of the day similar? These hormone in fact also fluctuate based on circadin rhythm.
The aim of the present study was to further investigate and confirm whether concentrations of copeptin, leptin, PAI-1, C-peptide, and insulin change before and after a 4-week dietary intervention with either SSRD or a low FODMAP diet in patients with IBS. Why did you decide to evaluate this primary aim and not whether these hormones are at least partially responsible for IBS symptoms?
Round 2
Reviewer 1 Report
Comments and Suggestions for Authors
Thepaper has been adequately revised accoridng to suggestions. I believe it is ready for acceptance.
Reviewer 2 Report
Comments and Suggestions for Authors
In consideration of the comprehensive evaluations and insightful analytical reasoning presented throughout this review, I am thoroughly convinced that the content and quality of the work are exceptionally well-prepared and refined. The depth of analysis, coupled with the clarity of expression, demonstrates a significant contribution to the field. Therefore, I firmly believe that this review is ideally suited for immediate publication. Its timely dissemination would not only enrich the existing body of knowledge but provides novel insights into how macronutrient modificationimpacts circulating hormone levels such as leptin and PAI-1 in IBS patients, thus advancingunderstanding of the molecular pathophysiology of IBS and its metabolic comorbidities.
Reviewer 4 Report
Comments and Suggestions for Authors
Thank you for your thorough revisions. The manuscript has improved significantly in terms of clarity and overall presentation. The primary aim of the article is now clear and understandable. No further major changes are required.